# Association between Prepartum Alerts Generated Using a Commercial Monitoring System and Health and Production Outcomes in Multiparous Dairy Cows in Five UK Herds

**DOI:** 10.3390/ani13203235

**Published:** 2023-10-17

**Authors:** John Cook

**Affiliations:** World Wide Sires, Yew Tree House, Carlisle, Cumbria CA1 3DP, UK; john.cook14@btinternet.com

**Keywords:** early lactation, milk yield, postpartum disease, transition alert, transition

## Abstract

**Simple Summary:**

Early lactation health disorders and their treatment are an important cause of welfare issues and production losses in dairy herds. These losses could be mitigated if cows at greater risk of health problems can be identified prior to calving. Cow mounted devices designed to monitor certain aspects of cow behavior may offer the opportunity to predict prior to calving which cows are likely to be at greater risk post calving by providing an alert based on each cow’s behavior. This study attempted to determine the level and accuracy of alerts provided by one such monitoring system. Results revealed an alert level of 38.7% with an accuracy of prediction of 62.5%. These findings reflect the level of early health disorders observed in commercial dairy herds. Further research is required to improve the accuracy of prediction for cows receiving treatments for ill health and to develop effective measures to prevent the occurrence of those treatments.

**Abstract:**

Identifying cows that are at greater risk for disease prior to calving would be a valuable addition to transition management. Prior to the commercial release of software features in an automated behavioral monitoring system, designed to identify cows in the dry period at greater risk of disease postpartum, a retrospective analysis was carried out in five dairy herds to evaluate whether the software could identify prepartum cows that subsequently received health treatments postpartum and whether prepartum alerts (transition alerts) are associated with a reduction in milk production in the subsequent lactation. Herd management and production records were analyzed for cows receiving treatment in the first 21 d of lactation (days in milk, DIM) for clinical mastitis, reproductive tract disease (metritis, retained fetal membranes), metabolic disease (hypocalcemia, ketosis and displaced abomasum) and for cows exiting the herd by 60 DIM. Data was gathered for 986 cows, 382 (38.7%) of which received a transition alert and 604 (61.3%) that did not. During the first 21 DIM 312 (31.6%) cows went on to receive a disease treatment, of these 51.9% (*n* = 162/312) were transition alert cows and 48.1% (*n* = 150/312) non-transition alert cows, while 8.6% (*n* = 33/382) alert cows exited the herd by 60 DIM compared to 4.8% (*n* = 29/604) of cows that did not receive an alert. A cow receiving a transition alert (OR = 1.76, 95% confidence interval (CI) = 1.27–2.44) and increasing parity (OR = 2.03, 95% CI = 1.44–2.86) were both associated with increased risk of receiving a disease treatment in the first 21 DIM. The occurrence of a transition alert was negatively associated with both week 4 milk yield (daily average yield in fourth week of lactation) and predicted 305 d yield. Transition alerts correctly predicted 62.5% (95% CI: 59.3–65.5) of treatments with a sensitivity of 42.4% (95% CI: 37.4–45.5) and a specificity of 75.2% (95% CI: 71.5–78.6). Associations were identified between postpartum health and production outcomes and prepartum behavioral measures from an automated activity monitoring system.

## 1. Introduction

Automated activity and behavior monitoring systems for the detection of estrus first came into use during the 1990s [1]. Since then, their use has expanded beyond estrus detection and their use has become increasingly common in dairy herds in many parts of the world. These systems provide the ability to remotely gather large amounts of real time data including activity, time spent ruminating, time spent eating, time spent lying, temperature of the ear skin, rumen pH, reticulo-ruminal temperature and cow location. Several leg, neck and ear mounted devices [2] as well as reticulo-rumen boluses [3] and camera-based systems [4] are commercially available. The behavioral and physiological data gathered by such systems can then be used either singly or in combination to identify changes indicative of poor health and generate actionable alerts that supplement visual observations for signs of ill health and assist in improving individual cow management.

Several studies have validated the accuracy of wearable devices across a range of behaviors [5,6] and linked changes in those behaviors to estrus [7], parturition [8,9,10], ill-health [11,12,13], calf management [14] and future reproductive success [15,16].

The risk of metabolic and infectious disease is greatest during the immediate postpartum period [17,18,19]. The majority (>90%) [20] of production diseases occur during the transition period with 50% or more of dairy cows becoming ill in the first 60 d of lactation (days in milk, DIM) often with multiple disorders [21]. Postpartum disease is often subclinical or so mild as to go unnoticed for extended periods of time despite having considerable effects on the wellbeing and future productivity of the cow [22,23,24]. Automated activity and behavior monitoring systems may offer opportunities for detection of postpartum disorders earlier than traditional clinical diagnosis by farm personnel [12].

Beauchemin [25] recently reviewed the factors influencing the feeding and rumination behavior of dairy cows and noted that, whilst the positive relationship between dry matter intake (DMI) and rumination is weak, most likely due to the effects of diet composition, the stronger positive relationship between eating time and DMI could be useful in estimating individual cows’ feed intake especially if combined with other information. Huzzey et al. also [26] studied changes in feeding behavior pre- and postpartum ultimately linking feed time and dry matter intake (DMI) with the occurrence of metritis [27]. A gradual decline in feed intake begins at around 3 weeks prior to calving [28] and precedes a much greater reduction of 30% 5 to 7 days prepartum [29,30]. Daily rumination time diminishes prepartum and reaches a nadir on the day of calving [31] or on the following day [32], but these changes are subject to large individual variation [31,33]. Postpartum recovery in daily rumination time to stable prepartum levels occurs between 7 and 15 days [31,34]. Despite these limitations, monitoring of cows during the transition period including the monitoring of rumination and eating may aid in early detection of subclinical problems and thus timely initiation of treatment or management change. This is important as the transition period defined as the 3 weeks pre calving until 3 weeks post calving [30] is the time when most health disorders occur [24,35] with substantial negative impacts upon a cow’s wellbeing and profitability. In this context Nebel and French [36] were able to use a combination of eating and rumination times to predict daily dry matter intake in pre- and postpartum cows. Twenty-six multiparous cows were fitted with ear-mounted sensors to record eating time, rumination time, activity, high activity, resting time and skin ear temperature, and assigned to electronic feeding stations capable of measuring daily feed intake and feeding times. 

Data from this study showed a high correlation between pre- and postpartum actual and predicted dry matter intake (0.66 and 0.74 respectively) and has since been modified for commercial use and incorporated into commercial software to create a daily ‘transition alert’ list of cows whose combined rumination and eating times indicate that their DMI is potentially predisposing them to a greater risk of disease postpartum. The system considers cows are eligible to receive transition alerts if the cow’s reproductive status is declared as ‘dry’ and is in the time period commencing 50 days prior to expected calving date. The objectives of this study were to validate the capability of the transition alert to predict the likelihood of disease treatments and culling occurring in early lactation as well as the association of a transition alert with future milk production. The ability to predict, prior to calving, which cows are likely to experience ill health in early lactation may offer future opportunities to develop preventive measures which can be applied before calving, and so help avoid health and welfare problems in early lactation. It was hypothesized that a transition alert would be associated with the occurrence of early lactation treatments, culling and milk production in the subsequent lactation.

## 2. Materials and Methods

### 2.1. Herd Selection

Herds were selected from a customer database and were approached and invited to participate in the study if they were of a suitable size (>400 cows), and had used the CowManager Sensor (Agis Automatisering, Harmelen, The Netherlands,) system for estrus detection and identification of possible sick cows postpartum for at least 12 months prior to any data collection being conducted. This system uses accelerometers placed in an ear-mounted tag to record eating time, rumination time, activity, high activity, resting time and skin ear temperature at 15-minute intervals. By using accelerometers arranged in a three-dimensional axis to detect acceleration as a measure of movement, the system can characterize movement of the ears as being typical of a cow engaged in activities such as eating and rumination. Each herd also had an electronic herd management software program (Uniform-Agri, Taunton, UK) that was used to record all health and fertility events and allowed access to data. Herds were composed solely of Holstein cows, used AI as the sole method of breeding, practiced year-round calving and participated in monthly milk recording. Feeding management was similar in each herd with a single total mixed ration fed to the lactating cows whilst far-off- (from end of lactation until approximately 3 weeks before expected calving) and close-to-calving (from approximately 3 weeks before expected calving until calving) dry cows were each fed a separate total mixed ration. Rolling herd average 305 d milk production was 10,480 kg at the time of study commencement. A partial anionic supplementation strategy [37] was used in all herds to achieve a target of −5 to 0 mEq/100 g of DM in the diet fed to close-to-calving cows. All diets were formulated to meet the requirements or exceed minimal nutritional requirements [38]. Routine vaccinations were carried out in all herds against bovine viral diarrhea virus, infectious bovine rhinotracheitis virus and *Leptospira interrogans* serovar *hardjo* and *Leptospira borgpetersenii* serovar *hardjo*. Cows on each herd were scheduled to be dried off into a far-off dry cow group 47 to 53 d before the predicted calving date (227–233 d of gestation) and then moved to a close-to-calving group 21 to 27 d (253–259 d of gestation) before the predicted calving date. Each herd relied on the generation of health alerts by the automated monitoring system supplemented with only visual daily inspection of cows’ demeanor to identify possible sick cows postpartum. A cow was eligible to enter the study if she had been fitted with a device from at least 50 days prior to her calving date until 90 DIM of the subsequent lactation or until she left the herd, whichever was earlier. The device was not allowed to have been replaced during this time and each cow’s records had to indicate that each day her device was functioning correctly as indicated by her sensor status in the automated monitoring system software.

### 2.2. Data Collection and Analysis

Prior to the commercial release and availability of the transition alert feature of the automated monitoring system the prepartum data for every cow calving between 1st June 2020 and 30th November 2020 was inspected retrospectively using the same algorithm that was to be released as part of the commercially available system and used to identify cows that would have been identified as transition alert cows had the system been available for on-farm use at that time. Briefly, this algorithm generates an alert for any cow whose daily combined eating and rumination time falls below a threshold of the herd mean for that day, minus one standard deviation. Alerts are generated for cows designated as dry and during a period of 50 days prior to the expected calving date. It should be noted that as this study was conducted retrospectively the dates of calving for each cow were the actual dates of calving and not those predicted from the software as would be the case in a prospective study design. Cows having 1 or more alerts prior to calving were categorized as having a transition alert (yes or no). Cows with alerts generated within 3 d of the date of calving were excluded from the analysis to avoid any potential confounding effects of calving. Herd management software records were also examined, and all details of any treatments administered to cows calving between 1st June 2020 and 30th November 2020 were extracted and entered into a spreadsheet (Excel, Microsoft Corp.). For each cow the unique identity, date of calving, lactation number and date of all health events, including details of all medical treatments administered in the first 21 DIM, were extracted from the Uniform-Agi software. Handwritten treatment records were also used when available so that the accuracy of the electronic records could be cross-checked. Each herd undertook milk sampling through National Milk Records or The Cattle Information Service on a monthly basis. Records of individual cow milk composition for each herd were electronically downloaded into DairyComp305 (Valley Agricultural Software, Tulare, CA, USA). From the DairyComp305 program the average daily milk weight produced by each cow during the fourth week of lactation (week 4 milk) was retrieved along with each cow’s projected 305-day milk production and previous lactation mature equivalent 305 d (305 d ME) milk yield. This step was necessary as the week 4 milk and 305 d ME are not available from the Uniform-Agri software, and daily milk weights were not available for each herd. Adopting this approach also ensured a standardized method of obtaining milk yield data. Average daily milk weight produced during the fourth week of lactation is calculated internally in DairyComp305 and has been used anecdotally to assess performance in early lactation. 

In the United Kingdom, all medicines administered to livestock species are required to be recorded in accordance with the Veterinary Medicines Regulations 2013, which includes a record of the unique identity of the animal receiving the medicine and the date of administration of the product. Data extraction and cross-checking was carried out in March 2021 to allow each cow to progress beyond 21 DIM for the collection of health records. Cows were categorized to allow for the effects of parity (lactation group) and season of calving. Lactation group indicates whether the cow is in her second, or third or more lactation. Group 2 contained cows only in second parity, and group 3 contained cows in third parity and above. Lactation 1 cows were not included in the study as they did not receive automated monitoring devices until after first calving. Season of calving distinguishes cows calving in summer (June, July and August) from those calving in autumn (September, October and November). As the health recording systems between farms were not consistent in defining each health event the reasons for treatments being administered to cows were also categorized by the likely body system involved as indicated from the data entry in the software. This resulted in treatments being placed into one of three categories defined as mastitis where the entry indicated the use of an intramammary preparation, or specifically mentioned mastitis, or the treatment of a specific quarter was mentioned, reproductive where the entry specifically mentioned metritis, uterine, vulval discharge, vaginal, or a combination of the wording retained placenta, membranes or cleansing, and metabolic if the entry mentioned ketosis, fatty liver, slow fever, off feed, displaced abomasum or milk fever. The terms cleansing and slow fever are colloquial terms used on these herds and were taken as meaning retained placenta and ketosis, respectively. Cows were then further categorized into those cows that received 1 or more treatments during the first 21 DIM and those that did not. Cows leaving the herd prior to 60 DIM as either sold or died were recorded as exits, and categorized yes or no.

### 2.3. Statistical Analysis

Statistical analysis was carried out using R version 4.0.4. [39]. Descriptive analysis was conducted using the summary function of the R coding system (package base). Multivariable logistic regression was used to model the odds of a cow receiving a treatment for a health event in the first 21 DIM (any combination of mastitis, reproductive tract disease or metabolic disease) using the glm function (family = binomial) of the R package stats. The fixed effects included in the model were lactation group, season of calving and occurrence of a transition alert. The farm where the animal was located was included as a fixed effect. The random effect of cow was considered in all models as nested within the transition alert group. Week 4 milk yield and predicted 305 d milk yields were modeled using general linear regression models using the following independent variables: lactation group, season of calving, occurrence of a transition alert, previous lactation 305 d ME milk production, occurrence of a treatment in the first 21 DIM and farm where the animal was located. To model time to exit event (sold/died) with right censoring at 60 DIM, a Cox proportional hazard model was fitted on lactation group, occurrence of a transition alert, treatment occurrence, season of calving and farm. Time to event data was also visualized using survival curves. The assumption of proportional hazards was evaluated both graphically and via statistical assessment using Schoenfeld residuals. To understand the main effect of covariates and obtain hazard ratios that would represent the survival curves a univariable Cox model containing each covariate was run for survival models. Survival curves were created for each covariate and tested for homogeneity using the Wilcoxon test. The DHARMA R package [40] was used to perform diagnostic checks and goodness-of-fit tests on the final models for treatment and milk production.

A receiver operating characteristic curve was constructed for the final treatment model and the area under the curve was calculated. All 2-way interactions between each covariates x transition alert were also tested in all models and final models were constructed using manual backward stepwise elimination. None of the interactions tested achieved a *p*-value ≤ 0.2 and were discarded from the final models. 

The record of a transition alert as a predictive tool for a cow receiving a disease treatment was assessed by calculating the sensitivity, specificity, and positive and negative predictive values along with the accuracy, and positive and negative likelihood of a transition alert cow receiving a treatment [41] (pp. 91–127).

Prior to data collection and analysis and assessment of the sample size necessary to detect a difference in treatment rates between groups with a 95% CI and 80% power [42] (pp. 33–53) was carried out. The pwr.2p2n.test function from the pwr R package [43] was used to determine the required sample size for the transition alert group to obtain the target power. Cohen’s h was set at 0.2 to ensure that a small difference [44] (pp. 179–213) in proportions of treated cows between groups could be detected. It was anticipated during the initial study design that from the herds recruited a study population of approximately 1000 cows would be available. As this method of generating alerts is new the difference in the proportion of treatment incidence between groups is unknown. The incidence of postpartum disease, however, has been estimated to vary between 30 and 50% [24,35], and this information was used to estimate the size of the transition alert group in our calculations and assumed that every occurrence of postpartum disease would be accompanied by an alert. Using this approach, a transition alert group of 272–322 cows would be necessary to detect a small difference in the proportion of cows treated for disease between groups.

## 3. Results

A total of 986 cows entered the study; 418 (42.4%) cows were in their second lactation and 568 (57.6%) in their third or greater lactation. First lactation animals were not included as no herds fitted animals about to enter their first lactation with automated monitoring devices until after their first calving. Month of calving ranged from June until November with 484 (49.1%) cows calving in the months of summer (June–August) and 502 (51.9%) cows calving in the autumn months (September–November).

Of the 986 cows that entered the study 382 (38.7%) received a transition alert and 312 (31.6%) cows subsequently received a treatment in the first 21 DIM for any combination of mastitis, reproductive tract disease and metabolic disease. Total treatments administered was 338 with 17 cows receiving a treatment for more than one reason, representing 5.3% of the treated cows. Overall, 42.4% (*n* = 162/382) of cows that received a transition alert also received a treatment during the first 21 DIM whereas 24.8% (*n* = 150/604) of cows that did not have a transition alert received a treatment. Herd exits by 60 days for the cows receiving a transition alert were 8.6% (*n* = 33/382) and 4.8% (*n* = 29/604) for those that did not.

Table 1 shows the results of the final logistic regression model for the occurrence of treatments in the first 21 d of lactation. Cows that received a transition alert were more likely (*p* < 0.001) to receive a treatment in the first 21 DIM than those that did not. Compared with second lactation cows those with three or more lactations were more likely to receive a disease treatment (*p* < 0.001). Cows on Farm 5 (*p* < 0.001) were also more likely to be treated for disease than those located on Farm 1.

Table 2 and Table 3 summarize the results of the final multivariable models on week 4 milk and predicted 305 d milk, respectively. Transition alerts were negatively associated with both week 4 milk (*p* < 0.01) and 305 d milk (*p* < 0.01). A cow receiving a treatment in the first 21 DIM had a negative association with both week 4 milk (*p* < 0.001) and 305 d milk (*p* < 0.001). Parity had a negative association with 305 d milk (*p* <0.01) but not with week 4 milk although there was a numerical negative association. There was an association between farm and both week 4 milk and 305 d milk. Cows from Farms 2 (*p* < 0.05) and 5 (*p* < 0.05) produced more week 4 milk than cows on Farm 1, while cows on Farms 2 (*p* < 0.001), 3 (*p* < 0.01), 4 (*p* < 0.001) and 5 (*p* < 0.05) produced more 305 d milk than cows on Farm 1. Greater previous lactation 305 d ME was associated with both higher week 4 milk (*p* < 0.001) and 305 d milk (*p* < 0.001). Calving in summer had a positive association with week 4 milk (*p* < 0.05) but not with 305 d milk.

Table 4 contains the results of the Cox proportional hazards regression model for cows exiting the herd by 60 DIM. Of the covariates tested, cows receiving a treatment within the first 21 DIM were more likely to leave the herd prior to 60 DIM (*p* < 0.01) (Figure 1B) as were cows in lactation group 3 (*p* < 0.05) (Figure 1C). Figure 1 displays the survival curves for each covariate tested in the Cox model.

The sensitivity, specificity, and positive and negative predictive values of a transition alert to predict the occurrence of treatment was 42.4% (95% CI: 37.4–47.5), 75.2% (95% CI: 71.5–78.6), 52% (95% CI: 46.2–57.6) and 67.4% (95% CI: 63.7–70.9), respectively. The positive and negative likelihood ratios were 1.71 (95% CI: 1.42–2.05) and 0.77 (0.70–0.85), respectively. A record of transition alert predicted a treatment with an accuracy of 62.5% (95% CI: 59.4–65.5).

## 4. Discussion

An association was identified between a transition alert occurring during the dry period and the likelihood that a cow would receive a disease treatment during the first 21 DIM of the next lactation. This is unsurprising as an association between changes in behavior prepartum and the occurrence of disease has already been established [23,26,27], and the generation of the transition alert is based on using combined daily eating and rumination times as measured by the automated monitoring system. This study, however, demonstrated poor sensitivity and only moderate specificity in the ability of a transition alert to identify which cows are likely candidates to receive treatment postpartum. This limits the utility of the alert as a screening process and a high standard of health monitoring is still required to be applied equally to all cows postpartum. Explanation for the poor performance of the occurrence of a transition alert as a predictor for future disease treatment may lie in the fact that social stressors such as stocking density, feed bunk space and social rank have also been shown to influence behavior in prepartum cows and alter the risk of being diagnosed particularly with uterine diseases as our understanding of the neuroendocrine and immune responses to stressful situations is limited [45]. 

As expected in this study a treated cow whether receiving a transition alert or not was associated with reduced milk production. These losses in production are well documented and result from the immediate withdrawal of milk from sale as well as future production losses along with other direct and indirect costs [46]. In addition to this expected milk loss, it was also observed that transition alert cows suffer an additional loss in milk production, which is independent of whether a cow received a treatment or not. Despite the low accuracy of the transition alert to predict a treatment occurring postpartum the combination of milk losses means that an alert system of low accuracy may still offer opportunities to avoid economic losses if cost effective interventions can be found. The cost benefits of such an approach are outside the scope of this study but it is an area that merits further study. Further explanation for the apparent poor ability to predict future disease treatments may lie in the fact that a threshold of 21 DIM was chosen for the recording of treatment events. The occurrence of postpartum health treatments is multifactorial, and most likely involves a combination of biological and management events that may occur either individually or in combination at various points in time both pre- and postpartum. By analyzing only treatment events occurring in the first 21 DIM cows were limited in their opportunity to become exposed to new postpartum risk factors. It is possible that some of the treatment events recorded were initiated either by factors occurring on the day of calving or in the immediate postpartum period when the cow is seeking to re-establish eating and ruminating behaviors [31,34,47] that are subject to large individual variations [31,33].

This study was conducted retrospectively, meaning there was no opportunity to standardize methods of disease diagnosis or recording between farms beyond the use of the automated monitoring system which may have produced a bias in the findings, and cases of disease and subsequent treatments may have been missed in cows that did not show health alerts postpartum. Standards of disease reporting and consistency of definitions vary greatly [48], and many cases may also go unreported as they are not considered severe enough or the clinical signs are interpreted subjectively and possibly incorrectly leading to an under-reporting of the true disease incidence [49]. However, in most herds of the size recruited into this study the financial consequences [50,51] (pp. 173–181) of failing to find and treat disease are well understood. Contact with these herds prior to data collection may also have alerted them and produced a bias in our findings. It was also apparent that users of automated systems of health monitoring are proactive in responding to cows that generate health alerts and often use non-specific treatment protocols to support these cows prior to the full development of clinical signs that would allow the cow to be categorized using consistent case definitions such as those described by Overton and Rapnicki [52]; this is a factor which may complicate future studies of the incidence of postpartum disease. However, it remains a major limitation of this study that it was not possible to apply consistent definitions of disease and standardization of treatments, and in this respect the findings should be treated with caution.

Although differences between groups in treatment events were detected it is possible that the assumptions used to calculate the required sample size were incorrect. Despite these limitations an increase of 71% in treatment incidence was detected between groups.

A difference in treatment incidence was observed between farms which may reflect differences in management systems and other areas influencing disease which were not addressed by this study. These factors induce social stress and alter the risk of disease [45,53,54]. It is also possible that differences arose due to differences in recording practices, and interpretation and classification of clinical signs [47] between herds although the overall incidence of treatments found in this study is in keeping with findings elsewhere [20,55].

Treatment incidence also rose with increasing parity which is unsurprising as the incidence of disease is known to rise as parity increases [56]. Season of calving was not shown to influence the likelihood of a cow needing a disease treatment which may seem surprising as heat stress has been shown to influence cow health [57] and it may be that in the relatively temperate climate of the United Kingdom other factors outweigh climatic effects.

This study observed an association between the occurrence of transition alert and both week 4 milk and 305 d milk production. This finding is of interest as it demonstrates that events prior to calving may influence milk production both in early lactation and throughout the whole of lactation. As expected, the occurrence of a treatment in the first 21 DIM was negatively associated with week 4 milk and 305 d milk production. It is also possible that many of the cows which recorded a transition alert in this study may have developed subclinical disease which was not identified in this study but may have contributed to the observed association between a transition alert and milk production. Subclinical disease has considerable consequences for milk production [18].

The effect of increasing parity showed no association with week 4 milk production but was negatively associated with 305 d milk production which may seem surprising. Previous lactation 305 d ME milk production was included in the analysis as a proxy to account for the effects of each cow’s individual merit for milk production which may have biased the outcomes for week 4 milk and 305 d milk. Previous lactation 305 d ME was positively associated with both week 4 milk production and 305 d milk production suggesting that the improvement in milk yield observed with increasing parity may be due to the survival of cows with better merit for milk production rather than the effect of increasing parity. An association between season of calving and week 4 milk production was observed but not 305 d milk production which contrasts with findings elsewhere [58]; however, this may be explained by the relatively temperate climate of the United Kingdom and the possible presence of other factors which were outside the scope of this study [52,53]. The effect of farm was shown to be associated with both week 4 milk production and 305 d milk production which should not be surprising as even though these herds were chosen to participate in this study based on some similarity of management system there is still great scope for variations in performance to arise from other factors [59]. Management standards and health care provided at and just after calving may also impact on subsequent milk production [60]. As expected, evidence was found that cows experiencing a treatment in the first 21 DIM were more likely to exit the herd as either sold or died by 60 DIM an association independent of whether a cow experienced a transition alert or not. It is possible that transition alert cows that do not experience a treatment may still experience subclinical problems postpartum that contribute to culling from the herd [18].

The findings of this study should be treated with caution as it was carried out on a small number of herds with better than average standards of management and so may not be representative of all herds.

## 5. Conclusions

The present study showed that the transition alert can be used to predict disease treatments that occur postpartum. However, test characteristics of the transition alert indicate a poor capability to accurately predict which cows are likely to receive a treatment postpartum and does not remove the need for good health screening postpartum. Preventative measures are justified to avoid both disease treatment in early lactation and early removal of cows from the herd. Additional milk production losses occur in transition alert cows beyond those due to cows receiving disease treatments. Further work is justified to both replicate the findings of this study and to improve the ability of the transition alert to predict cows at risk of disease treatment. Work to identify effective measures to prevent the occurrence of transition alerts is also justified. 

## Figures and Tables

**Figure 1 animals-13-03235-f001:**
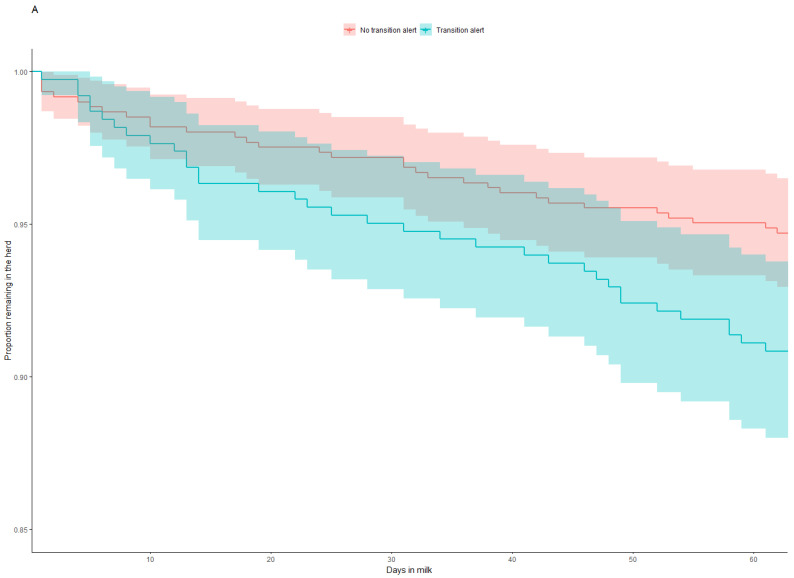
Survival curves for time to leave the study by either death or culling to 60 DIM by (**A**) transition alert group, (**B**) treatment occurrence in the first 21 DIM, (**C**) lactation group, (**D**) season of calving and (**E**) farm where the cow was located. Shaded areas represent 95% confidence intervals.

**Table 1 animals-13-03235-t001:** Final logistic regression model for the occurrence of treatments in the first 21 d of lactation ^a^.

				95% CI ^b^		
Model Term	Estimate	Standard Error	Odds Ratio	Lower	Upper	*p*-Value
Constant	−1.91	0.22				
Transition alert status						
No			Referent			
Yes	0.56	0.17	1.76	1.27	2.43	<0.001
Farm 1			Referent			
Farm 2	−0.47	0.27	0.62	0.37	1.06	0.08
Farm 3	−0.36	0.25	0.70	0.43	1.14	0.15
Farm 4	−0.01	0.26	1.00	0.60	1.63	0.96
Farm 5	1.13	0.25	3.08	1.90	5.01	<0.001
Lactation group						
2			Referent			
3+	0.71	0.17	2.03	1.44	2.86	<0.001
Season (summer vs. autumn)						
Autumn ^c^			Referent			
Summer ^d^	0.04	0.16	1.04	0.76	1.42	0.82

^a^ For this model, the area under the receiver operating characteristic curve is 0.70. ^b^ 95% CI, 95% confidence interval. ^c^ Autumn, season of calving, September, October and November. ^d^ Summer, season of calving, June, July and August.

**Table 2 animals-13-03235-t002:** Final linear regression model for week 4 milk yield.

Model Term	Estimate	Standard Error	*p*-Value
Intercept	31.96	1.94	<0.001
Transition alert status			
No		Referent	
Yes	−1.88	0.60	<0.01
Treatment in first 21 DIM			
No		Referent	
Yes	−4.73	0.70	<0.001
Previous lactation 305 d ME	0.001	0.0002	<0.001
Farm 1		Referent	
Farm 2	1.88	0.89	<0.05
Farm 3	−0.46	0.86	0.60
Farm 4	1.19	0.91	0.19
Farm 5	1.96	0.96	<0.05
Parity			
2		Referent	
3+	−0.55	0.70	0.43
Season (summer vs. autumn)			
Autumn ^a^		Referent	
Summer ^b^	1.19	0.55	<0.05

^a^ Autumn, season of calving, September, October and November. ^b^ Summer, season of calving, June, July and August.

**Table 3 animals-13-03235-t003:** Final linear regression model for predicted 305 d milk production.

Model Term	Estimate	Standard Error	*p*-Value
Intercept	5865.1	412.84	<0.001
Transition alert status			
No		Referent	
Yes	−344.2	130.17	<0.01
Treatment in first 21 DIM			
No		Referent	
Yes	−654.0	151.21	<0.001
Previous lactation 305 d ME	0.5	0.04	<0.001
Farm 1		Referent	
Farm 2	819.5	192.75	<0.001
Farm 3	488.3	187.31	<0.01
Farm 4	1088.8	197.23	<0.001
Farm 5	463.4	208.06	<0.05
Parity			
2		Referent	
3+	−675.5	150.41	<0.001
Season (summer vs. autumn)			
Autumn ^a^		Referent	
Summer ^b^	−173.8	120.15	0.15

^a^ Autumn, season of calving, September, October and November. ^b^ Summer, season of calving, June, July and August.

**Table 4 animals-13-03235-t004:** Regression coefficients, hazard ratios (HR), 95% confidence intervals for the hazard ratios (in brackets) and *p*-values for the Cox proportional hazards regression model.

	Exit as Died/Sold by 60 DIM
Transition alert status (No vs. Yes)	Yes: 0.37 HR: 1.45 (0.85–2.48) *p* = 0.17
Lactation group (2nd vs. 3 or more)	3 or more: 0.63 HR: 1.88 (1.02–3.47) *p* < 0.05
Farm (2 vs. 1)	2:−0.48 HR: 0.62 (0.27–1.44) *p* = 0.26
Farm (3 vs. 1)	3:−0.70 HR: 2.00 (0.22–1.13) *p* = 0.10
Farm (4 vs. 1)	4: 0.91 HR: 1.10 (0.53–2.26) *p* = 0.80
Farm (5 vs. 1)	5:−0.27 HR: 0.76 (0.35–1.67) *p* = 0.50
Season (summer vs. autumn)	Summer: 0.41 HR: 1.51 (0.91–2.52) *p* = 0.12
Any treatment in first 21 DIM (No vs. Yes)	Yes: 0.83 HR: 2.30 (1.35–3.93) *p <* 0.01

## Data Availability

The data presented in this study are available on request from the corresponding author.

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
