# Peer review of "Association between Prepartum Alerts Generated Using a Commercial Monitoring System and Health and Production Outcomes in Multiparous Dairy Cows in Five UK Herds"

_animals, 2023, doi:10.3390/ani13203235_

Round 1

Reviewer 1 Report

Title: Association between Prepartum Alerts Generated Using a ComMercial Monitoring System and Health and Production  Outcomes in Multiparous Dairy Cows in Five UK Herds..

Manuscript is written so general and It is not clear. Main problem is in association between prepartum alerts (eating time, rumination time, activity, high activity, resting time, and skin ear temperature at 15-minute intervals) with postpartal  metabolic, reproduction and infection detected disorders (mastitis, metritis, uterine, vulval discharge, vaginal, or a combination of the wording retained placenta, membranes, or cleansing and metabolic as ketosis, fatty liver, slow fever, off feed, displaced abomasum, or milk fever anmd othhers) which increase therapeutic cost and decrease milk production in the herds.

I do not see  changes in prepartum detected parameters alerts with final results.

Finnaly, manuscript is not clear and It  is not based on the strong  scientific facts and associations.

This manuscript do not satisfies a high level of Journal (animals), and I suggest to author to submit manuscript in dairy industry journal.

Author Response

Comments from the reviewer are confusing and difficult to understand. Particularly the comment that they do not see changes in prepartum detected parameters alerts with final results. nothing was actually measured prepartum except recording whether a cow received an alert yes or no and that is clearly reported.

Reviewer 2 Report

This study considered retrospective data from five herds (each having >400 Holstein cows) all of which used a specific sensor-based system for oestrus detection and identification of sick cows for at least 1 year before data collection. Cows could enter the study if fitted with the sensor at least 50 days before calving and until the subsequent 90 DIM (or until leaving the herd if earlier). Pre-partum data for cows calving from June 1st to Nov 30th 2020 were collected retrospectively to examine a new transition alert feature (algorithm) of the system prior to commercial release. The algorithm generates an alert if the daily combined eating and rumination time falls below the herd mean for that day minus one standard deviation. A number of different responses were examined for association with transition alert (amongst other fixed effects) by use of sound statistical analyses: treatment within 21 DIM post-partum, week 4 milk yield, predicted 305-d milk yield, time to exit herd events (censored at 60 DIM). The use of the transition alert as a predictive tool for disease treatment early post-partum was also examined by usual measures. Association was found between the transition alert pre-partum and treatment the first 21 DIM post-partum but the predictive performance was not too impressive.

It is likely that some readers/reviewers will say that the statistical analysis section is a bit (too) long; however, as a statistician I find it to be of suitable length with the details needed to replicate/reproduce the results. I have a few comments though.

A general comment to the association analyses: If you want to think of the results as being general, would it then not be obvious to include Farm as a random effect? By omitting Farm from the fixed effects, you also would not have to deal with the multiple comparison (which you have not done) among herds. On the other hand, you point out the fact that the herds in the present study may not be fully representative. I am fine with the choice of having Farm as a fixed effect but some readers may wonder so perhaps you could touch upon the choice in the discussion or in the statistical analysis section.

Introduction

P2-L73: you use DMI before defining it (e.g., P2-L68)

P2-L86: ‘Data from this study’ ... I suppose you now start describing the present study!? I think starting on a new paragraph would make this clearer.

Materials and Methods

P4-L106: I think you would like to delete the first ‘mounted’

P5-L199-200: Although a linear normal model (normal linear regression), is also a generalized linear model, you likely meant ‘generalized’ to be “general” ... and if not you will need to specify the family (which from Table 2 and 3 appears to be Gaussian, i.e. a normal linear regression).

P5-L203: I suggest adding what the event is. According Table 4 and Figure 1 it is exiting (died/sold) by 60 DIM. You for example change to something like ‘To model time to exit event (died/sold) with right censoring at 60 DIM, a Cox proportional ...’

P6-L207-210: Why not just present the estimated survival curves from the multivariable model? (see comments to Figure 1 below)

Results

P6-L248: ‘TA’ has not been defined (though it likely stands for “transition alert”). Do you used it elsewhere?

Table 1: I do not think that I would normally show the ‘Constant’ term for a logistic regression – but it does not hurt. However, what would your interpretation in terms of the odds ratio (exp of the estimate) be? A suggestion could perhaps be to show only the estimate and the standard error for this parameter. As a perhaps personal preference I would include the zero before .001 in ‘< .001’, i.e. write “< 0.001” ... you actually do so in the text ... well, actually you also do this in Table 2 and 3. Regarding the effect of Farm: you have included five farms and thus there are 10 possible comparisons – the results of which (after correction for multiple testing) could be indicated by letters. It seems likely that Farm 5 differs significantly not only from Farm 1 (the reference in your analysis) but also from the other three farms.

P6-L256-257: see above comment regarding Farm 5.

Table 2: Again, personal preferences ... and I do not know what the journal prefers ... I prefer p-values that are less the chosen level of significance (often 0.05) but above 0.001 to be presented with three decimals instead of using ‘< 0.01’ when 0.01 > p >= 0.001. I know some journals (and authors) prefer using precisely two decimals (except for the < 0.001) but this essentially means that the non-significant p-values above 0.10 are shown with two significant figures whereas the interesting significant p-values are shown with only one. By doing this you would (maybe) also not need to us ‘< 0.05’ (which is used, I suppose, for 0.05 > p >= 0.045). The estimate for the ‘Previous lactation 305-d ME’ could be shown per 1000 ME instead of per ME. After correction for multiple testing I doubt any of the farms differs from each other ... again, there are more comparisons to make than those with Farm 1.

Table 3: Same comment regarding previous 305-d ME as above. I think one decimal is enough for the estimates (and then I prefer having one more digit for standard errors than for the means). Actually in my opinion showing means as integers would also be fine in this case (and then one decimal for standard errors).

Table 4: Why not use the same structure as used in Table 1? There are six more comparisons among Farms than those shown (and adjusting for multiple testing would also be relevant). In this table you have shown a p-value < 0.01 and > 0.001 (P = 0.002) ... why here but not in the other tables? In addition, why not show the third decimal for the p-value = 0.05? Alternatively, use the ‘< 0.05’ that you used in Table 3?

Figure 1: confidence bands would be nice (at least for panels A, B, C and D). I wonder why you chose to cut the y-axis at 0.7 ... why not higher – none of the proportions are even close to getting down to 0.7 and as such the axis should go down to 0 if you want to present the likeliness of the events. You could also choose to present Nelson-Aalen cumulative hazards instead. If the curves were estimates from a Cox regression, I would call it survival curves. Kaplan-Meier estimates is a non-parametric method of obtaining the survival function (survival curves) but those from the Cox model are not Kaplan-Meier estimates. Actually, plotting the estimated survival curves (from the Cox model) along with the Kaplan-Meier estimated curves can be used as a model diagnostics tool. If the objective is to present graphically the effects of the covariates, why not use the estimated curves from the multivariable model?

P9-L316 and L317: See the comment to Table 4 regarding the p-values.

P12-L358: have a look at the parenthesis ‘(69.5 – 84.5)’ ... did you coincidently multiply by 100? More structure of the two confidence intervals presented here is different from the structure used in L356, L357 and L359.

Discussion

P13-L362: ‘occurring dry the period’ ... something is wrong with this sentence

Author Response

Many thanks for your time to review this manuscript.

There is a general comment about the use of Farm as a fixed effect and we would like it noted that we did consider including Farm as a random effect and indeed carried out an analysis on that basis. Unfortunately we also received conflicting advice on the matter from people with specialist knowledge and a decision was made to stick with our original plan as the alternative did not alter any of the significant effects or effect sizes substantially.

From reading the review we think the reviewer recognises there are different ways of reporting and displaying data which are not necessarily right or wrong indeed our own experience suggests that when consulting 2 statisticians it is possible to get multiple answers none of which is incorrect.

Line 73. DMI has now been defined at line 74.

Line 86. A new paragraph now starts at line 91.

Line 106. Thank you, the extra mounted has been deleted. See line 113.

Line 199-200 Has been amended to 'general' and is locate at line 209-210 although the family used was the default Gaussian.

Line 203. text has been amended as suggested by the reviewer and is located at line 213.

Line 207-210 and Figure 1. We agree we could have chosen to present the curves from the Cox model but chose not too as it was felt the KM charts better displayed the differences in effects between variables. The KM charts were also used in our exploratory work up to visualise effects prior to developing the Cox model so they seemed more appropriate. A cut off of 0.7 was chosen simply to provide charts that fitted best to the template of the page for the reader. As the reviewer points out the confidence bands may differ from the model estimates and so may confuse the reader beyond the visual information we wished to convey. In the revised manuscript a cut off of 0.8 is now used and 95% CI have been added to the KM graphs which are now described as 'survival curves' at lines 216, 218, 219 and 362. Test has been added to the figure to indicate the shade areas show the confidence intervals.

Line 248 and elsewhere TA has been replace with 'transition alert'.

Table 1. and Line 256-257. As suggested only the estimate and standard error remain for the constant term (line 269). Again in our work up of this data we received conflicting advice on this matter and for the record agree with this reviewer.

As regards multiple comparison between farms there were significant differences which are not presented in the manuscript, however it was not the intention of this study to explore differences between farms. Rather we intended to explore if the occurrence of alerts was useful across farms in general.

Table 2 has been reviewed and amended for consistency in presenting P values.

Table 3. Amended as suggested.

Table 4 was presented in this format on the basis of external advice received. P values have been amended for consistency.

Line 358 the multiplication error has been corrected. the amendments are at lines 367 - 372.

Line 362 the word 'the' has been changed to 'a' at line 375.

Reviewer 3 Report

Overall the manuscript is well written. I only have a few comments and suggestions for the author to consider.

Abstract:
Line 26: It might be helpful to define what reproductive tract diseases and metabolic disease are or just include the major diseases that are included in these.

Line 103:  I assume that these herds are in the UK?  Also, it would be nice to include some basic information on these herds.  Average Production, fertility, lactation number of the herds.  Especially production.

Line 190:  What were the incidence rates of the diseases in the that you used for these herds?  Or the categories.

Stat Analysis:
Line 191: I am perplexed why you might not include rumination or activity in the statistical model as a covariate.  I would think that if you included at least rumination, maybe the results would be different. I am not sure.  Rumination goes down, and possibly activity and you might be able to predict the transition alert.   

I know CowManager also include eating which might be interesting as well.  If these are not inlcluded in the statistical model, there should be at least some of this included in the discussion based on these factors.

Author Response

Firstly many thanks to the reviewer for undertaking this work.

Line 26 text has been added to identify the diseases included in each category.

Line 103.  It is stated in the article title that these herds are located in the UK. At line 124-125 rolling herd average 305 d milk production has been added to the text.

Line 190. This information was omitted from the manuscript as incidence of each condition varied vastly between herds making them largely meaningless and added little to the understanding of the paper and its objectives which were to explore the utility of the alert to predict cows requiring treatment, not to explore diseases incidence. Further more the study was not looking at disease incidence but the incidence of treatment for disease as recognised by a farmer, the two may not be the same. For example only 16 cows received a treatment for reproductive issues on one farm while another recorded 50 with differing proportions of treated cows receiving alerts.

Line 191. Rumination time alone is known not to be a good predictor of future disease and the work of Nebel and French 2019 demonstrated that a combination of rumination and eating provided the best prediction of a cows dry matter intake and proprietary data from AGIS and elsewhere indicates that rumination alone is not a good predictor of future disease in a close up cow. for these reasons it was excluded as a predictor in this study. It is included as the alert yes/no. the alert is based on cows that fall below a threshold of combined rumination and eating time found at line 148-149. We did not explore this route in our analysis so cannot say if there would have been issues of confounding. Our objective was also to explore the utility of the commercial alert as a predictive tool and not look at a single behaviour. The alert is what is being marketed by the company in question and it is useful for farms to have published information available to evaluate if they are considering purchasing the device. As we did not investigate the individual effect rumination and eating it seems inappropriate to include it in our discussion which should surely seek to discuss the results of the present study.

Reviewer 4 Report

General Comments

This paper effectively highlights the significance of early lactation health disorders in dairy herds

While the abstract provides a concise overview, it could be slightly expanded to include a sentence or two about the potential broader implications of these findings for the dairy industry.

Remember that your current text is quite detailed, which is great for conveying specific information. However, in academic or scientific writing, it's important to find a balance between detail and readability to ensure that your audience can grasp the key points easily.

Particular Comments

1.    Introduction: It would be helpful to include a brief statement about why identifying cows at greater risk before calving is important. This could provide context for readers who might not be familiar with the subject.

2.    Transition Management Importance: Emphasize the importance of transition management in dairy farming. Transition management refers to the period around calving when cows are particularly vulnerable to health issues. This will help readers understand the relevance of identifying high-risk cows during this phase.

3.    Methodology Explanation: Consider providing a bit more detail about how the behavioral monitoring system functions. This will aid in understanding how the alerts are generated and how cow behavior is being tracked.

4.    Results Clarity: The results are presented clearly, showing the percentage of cows receiving transition alerts and their subsequent health outcomes. To enhance clarity, you could provide a summary table that includes these percentages along with the key health issues (clinical mastitis, reproductive tract disease, metabolic disease) and their respective percentages for alert and non-alert cows.

5.    Language and Structure: Ensure consistency in terminology. For instance, consider using "prepartum" consistently instead of "pre-partum" for uniformity. Additionally, break down lengthy sentences into smaller ones to enhance readability.

6.    Citations: If there are any specific references or studies that have influenced this work, consider citing them within the text. This can provide additional context and support for your findings.

7.    I would like to suggest the following references in their revised version.

(i)            Sumi, K.; Maw, S.Z.; Zin, T.T.; Tin, P.; Kobayashi, I.; Horii, Y. Activity-Integrated Hidden Markov Model to Predict Calving Time. Animals 202111, 385. https://doi.org/10.3390/ani11020385

(ii)          Maw, S.Z.; Zin, T.T.; Tin, P.; Kobayashi, I.; Horii, Y. An Absorbing Markov Chain Model to Predict Dairy Cow Calving Time. Sensors 202121, 6490. https://doi.org/10.3390/s21196490

    Language and Structure: Ensure consistency in terminology. For instance, consider using "prepartum" consistently instead of "pre-partum" for uniformity. Additionally, break down lengthy sentences into smaller ones to enhance readability.

Author Response

Firstly the author would like to thank the reviewer for their time on this manuscript.

Addressing points 1 and 2 additional text has been inserted at lines 82-85 and 100-103.

'This is important as the transition period defined as the 3 weeks pre calving until 3 weeks post calving [30] is the time when most health disorders occur [24, 35] with substantial negative impacts upon a cow’s wellbeing and profitability.'

'The ability to predict prior to calving, which cows are likely to experience ill health in early lactation may offer future opportunities to develop preventive measures which can be applied before calving and so help avoid health and welfare problems in early lactation.'

Point 3. At lines 115-117 additional text has been inserted.

'By using accelerometers arranged in a three-dimensional axis to detect acceleration as a measure of movement it is possible to characterize movement of the ears as being typical of a cow engaged activities such as eating and rumination.'

Regarding point 4 this data was omitted from the manuscript as there was a large amount of variation in disease occurrence between farms for different categories, for example one farm had only 2 cases of mastitis with 100% being alerts whilst another had 9 cases with a much fewer proportion of alerts. It was felt this information was meaning less and added little to the understanding of the paper.

Point 5. The manuscript has been reviewed and revised as suggested. Examples at lines 477 and 479.

Point 6 and 7 additional references have been added as suggested. Authors are grateful to the reviewer for bringing these to attention. The work of Nebel and French is cited at line 35 and is the work upon which the transition alert is based and prompted the work for this study.

Round 2

Reviewer 1 Report

I stay to my previous opinion.

Author Response

We have no further comment